# Evaluating Differentiation Status of Mesenchymal Stem Cells by Label-Free Microscopy System and Machine Learning

**DOI:** 10.3390/cells12111524

**Published:** 2023-05-31

**Authors:** Yawei Kong, Jianpeng Ao, Qiushu Chen, Wenhua Su, Yinping Zhao, Yiyan Fei, Jiong Ma, Minbiao Ji, Lan Mi

**Affiliations:** 1Key Laboratory of Micro and Nano Photonic Structures (Ministry of Education), Department of Optical Science and Engineering, Shanghai Engineering Research Center of Ultra-Precision Optical Manufacturing, School of Information Science and Technology, Fudan University, Shanghai 200433, China; 18110720005@fudan.edu.cn (Y.K.); 21110720091@fudan.edu.cn (Q.C.); 20110720012@fudan.edu.cn (W.S.); fyy@fudan.edu.cn (Y.F.); jiongma@fudan.edu.cn (J.M.); 2Department of Physics, Fudan University, Shanghai 200433, China; 20110190072@fudan.edu.cn; 3Human Phenome Institute, Fudan University, Shanghai 200433, China; 19110860079@fudan.edu.cn; 4Institute of Biomedical Engineering and Technology, Academy for Engineering and Technology, Fudan University, Shanghai 200433, China; 5Shanghai Engineering Research Center of Industrial Microorganisms, The Multiscale Research Institute of Complex Systems (MRICS), School of Life Sciences, Fudan University, Shanghai 200433, China

**Keywords:** MSCs, label-free, FLIM, SRS, machine learning

## Abstract

Mesenchymal stem cells (MSCs) play a crucial role in tissue engineering, as their differentiation status directly affects the quality of the final cultured tissue, which is critical to the success of transplantation therapy. Furthermore, the precise control of MSC differentiation is essential for stem cell therapy in clinical settings, as low-purity stem cells can lead to tumorigenic problems. Therefore, to address the heterogeneity of MSCs during their differentiation into adipogenic or osteogenic lineages, numerous label-free microscopic images were acquired using fluorescence lifetime imaging microscopy (FLIM) and stimulated Raman scattering (SRS), and an automated evaluation model for the differentiation status of MSCs was built based on the K-means machine learning algorithm. The model is capable of highly sensitive analysis of individual cell differentiation status, so it has great potential for stem cell differentiation research.

## 1. Introduction

Mesenchymal stem cells (MSCs) are a type of stem cell that exist in various tissues of the human body, including bone marrow, adipose tissue, and the umbilical cord [1,2,3,4]. First discovered and isolated by Friedenstein in the 1960s [5], MSCs are able to self-renew, differentiate into multiple cell types, and secrete cytokines. Since they do not raise ethical concerns and have a wide range of clinical applications in areas such as plastic and reconstructive surgery, bone transplantation, and trauma treatment, MSCs have attracted significant attention [6,7,8,9,10]. In the fields of organ engineering and clinical stem cell therapy, the determination of differentiation status is essential. Implanting an undifferentiated cell count of 10^4^ in vivo can cause teratoma [11], and the number of differentiated cells that are implanted directly determines the effectiveness of clinical treatment. However, the heterogeneity of MSCs during the production process has gradually increased due to the unique donors, varying isolation methods, differences in culture conditions, freeze thawing, and lack of unified standards [12]. Additionally, there is significant heterogeneity in the proliferation and differentiation of MSCs, which is a major obstacle to their clinical application [13,14]. Currently, conventional detection methods such as immunofluorescence staining, chemical staining, Western blot, and real-time quantitative polymerase chain reaction all require labeling or lysis of cells, which are invasive and only applicable to cells that are no longer useful for other purposes after analysis. Because it is not possible to determine differentiation status in situ or without labeling, the clinical application of MSCs is severely limited. Therefore, label-free and noninvasive methods for detecting the differentiation status of cells are essential for MSC research.

The free and protein-bound states of the intracellular metabolic cofactor nicotinamide adenine dinucleotide (phosphate) (NAD(P)H) are closely related to cellular metabolism. Change in NADH at different levels of cell differentiation has been observed in various reports [15,16,17,18]. Hsu et al. performed Western blot analysis of mitochondrial respiration enzyme Complex I, which is closely related to NADH, in MSCs during adipogenic differentiation, and found a significant increase in the band intensity of the enzyme Complex I during differentiation, revealing an increase in the proportion of protein-bound NADH during adipogenic differentiation [15]. Guo et al. also performed Western blot analysis of mitochondrial respiration enzyme Complex I in MSCs during osteogenic differentiation and observed a significant increase in the band intensity of enzyme Complex I, leading to the conclusion that the proportion of protein-bound NADH increases during osteogenic differentiation [16]. Edgar et al. characterized the concentration of NADH using fluorescence intensity, and concluded that the concentration of protein-bound NADH increases during adipogenic differentiation, reflecting the shift in metabolism from glycolysis to oxidative phosphorylation during differentiation [17]. Rice et al. developed a method to quantify the concentration of fluorescent groups based on fluorescence intensity, and studied the changes in NAD(P)H concentration during adipogenic and osteogenic differentiation of MSCs using NAD(P)H autofluorescence, demonstrating that the intracellular NAD(P)H concentration gradually increases during adipogenic and osteogenic differentiation [18]. In addition, more studies have reported changes in NADH fluorescence lifetime as a reflection of cellular metabolic changes during differentiation [19,20,21]. Chen et al. reported the activation of oxidative phosphorylation during osteogenic differentiation of MSCs [19], and Meleshina et al. used FLIM to test the information on the proportion of protein-bound NAD(P)H during differentiation, and found that the increase in the proportion of protein-bound NAD(P)H can reflect the increase in the ratio of oxidative phosphorylation [20]. Guo et al. also obtained similar conclusions by testing the changes in NADH fluorescence lifetime during osteogenic differentiation of MSCs [21]. Therefore, using the fluorescence lifetime of NADH to characterize the metabolic changes during MSC differentiation is feasible. Based on the autofluorescence of NAD(P)H, fluorescence lifetime imaging microscopy (FLIM) has been used as a noninvasive detective method to intuitively reflect changes in cellular metabolic state and has been applied in the determination of MSC differentiation status. Guo et al. [16] used a two-photon FLIM system to collect NAD(P)H data during the differentiation of MSCs into osteoblasts and distinguished differentiation states at several time points through statistical analysis. Meleshina et al. [20,22] analyzed the metabolic trajectory of MSCs during differentiation into osteoblast, adipocyte, and chondroblast cells using a similar approach. Chakraborty et al. [23] analyzed metabolic changes in MSCs during osteogenic and adipogenic differentiation using FLIM technology.

Spontaneous Raman spectroscopy is a label-free analysis method that can provide information on specific molecules within cells and reveal molecular changes at the single-cell level. It has been applied to detect MSC differentiation. Lazarević et al. [24] used spontaneous Raman spectroscopy to reveal changes in proteins, lipids, and nucleic acids within MSCs during differentiation, and to discriminate between cells differentiated towards osteogenic, adipogenic, and chondrogenic lineages. Ravera et al. [25] used spontaneous Raman spectroscopy to study the chondrogenic differentiation of MSCs and evaluated their differentiation status at the single-cell level by analyzing spectral features of the nucleolus and cytoplasm. However, the application of spontaneous Raman spectroscopy is greatly limited by its weak signals, long integrated time, and difficulty in collecting complete spectra from individual cells. Stimulated Raman scattering (SRS) is another label-free technique, but unlike spontaneous Raman scattering, which is excited by two coherent lasers. Its signal is strong and with no non-resonant background, and allows for fast imaging [26]. Based on these advantages, SRS has been widely applied in the study of lipid changes within cells [27,28]. Both the energy metabolism and cell contents change significantly during MSCs differentiation. The energy metabolism of undifferentiated MSCs is mainly glycolytic. Upon osteogenic differentiation, the energy metabolism mainly shifts from glycolysis to oxidative phosphorylation, resulting in increased expression of alkaline phosphatase (ALP). Upon adipogenic differentiation, the energy metabolism change is similar to osteoblasts, but with increased accumulation of intracellular lipids [29]. There is little research on stem cell differentiation using the SRS technique.

Both FLIM and SRS can provide label-free microscopic images for cells. Due to the strong heterogeneity among cells during differentiation, statistical analysis of a large number of cells FLIM or SRS images can only show the average trend of metabolic changes or cell contents changes and cannot accurately reflect the differentiation state of individual cells. To investigate the individual cells, image segmentation and classification can be achieved using machine learning.

Machine learning methods have been widely applied to investigate the cellular morphology of MSCs [30,31]. Mota et al. obtained phase-contrast microscopy images of MSCs and employed machine learning methods to extract morphological features and classify cells, achieving accurate classification of low-density and medium-density MSCs [30]. Marklein et al. employed a high-content imaging system to acquire images of MSCs under various inflammatory cytokine interferences and studied the morphological features using machine learning methods, successfully identifying distinct morphological subgroups of MSCs [31]. Machine learning methods have also been extensively applied in metabolic research [32,33]. Bianchetti et al. employed machine learning methods in the investigation of metabolic alterations in the MCF-7 breast cancer cell line. They utilized the fluorescence lifetime information of NADH as features and achieved intracellular metabolic state detection at the pixel level [32]. Ji et al. collected FLIM data of NAD(P)H in cervical exfoliated cells and applied unsupervised clustering using machine learning methods to screening cervical cancer with higher sensitivity [33]. Machine learning methods combined with SRS have been used to provide rapid and accurate diagnosis of breast cancer by studying lipid/protein contents [34,35]. Liu et al. successfully achieved simulated rapid and automated intraoperative diagnosis by using SRS to detect lipid signals in gastric cancer tissue, combined with machine learning methods [34]. After obtaining lipid signals in laryngeal cancer tissue through SRS, Zhang et al. combined machine learning methods to achieve rapid intraoperative diagnosis [35]. Consequently, this work utilized machine learning algorithms to extract and evaluate multiple cellular features, including cellular morphology, NAD(P)H fluorescence lifetime, and lipid contents, from the FLIM and SRS images as previously mentioned. Furthermore, the research work on MSC differentiation requires labeling of each cell for supervised analysis, but it is difficult to obtain cell labels during the differentiation process. Thus, in this study, a more efficient unsupervised machine learning method to discriminate the differentiation status of stem cells was developed.

This article presents a new approach for evaluating the differentiation of MSCs into adipogenic and osteogenic lineages. FLIM and SRS data were collected to characterize the energy and substance metabolism of MSCs induced to differentiate into these two lineages. Single cells were obtained from high-density cell populations using a sliding window segmentation method, and the extracted feature data were input into K-means clustering. By implementing this unsupervised machine learning approach, the differentiation states of MSCs into adipogenic and osteogenic lineages were distinguished at the single-cell level in a label-free, non-invasive, and high-throughput manner, even under highly heterogeneous conditions. These results provide new insights for the clinical research of MSCs.

## 2. Materials and Methods

### 2.1. Preparation of Culture Medium

The MSC culture medium was prepared by mixing 44.5 mL basal medium (HUXMA-90062, Cyagen Biosciences, China) with 5 mL premium fetal bovine serum (FBSAD-01011-100, Cyagen Biosciences, China) and 0.5 mL penicillin–streptomycin solution (ATPS-10001-10, Cyagen Biosciences, China).

The adipogenic induction medium A contained 10% premium fetal bovine serum, 1 mM dexamethasone, 0.5 mM isobutylmethylxanthine (I106812-250mg, Aladdin, China), 50 μM indomethacin (I106885-5g, Aladdin, China), 10 μg/mL insulin (I302196, Aladdin, China), and 1% penicillin–streptomycin solution. The adipogenic induction medium B contained 10% premium fetal bovine serum, 10 μg/mL insulin, and 1% penicillin–streptomycin solution.

The osteogenic induction medium contained 10% premium fetal bovine serum, 100 nM dexamethasone (D137736-1g, Aladdin, China), 10 mM β-glycerol phosphate (D106347, Aladdin, China), 50 μM ascorbic acid (A103539-500g, Aladdin, China), and 1% penicillin–streptomycin solution.

All the prepared media was wrapped in aluminum foil and stored at 4°C for later use.

### 2.2. Culture and Induction of MSCs

MSCs at freezing passage 0 (P0) were obtained from the Cell Bank of the Chinese Academy of Sciences (Shanghai, China). The cryovial was removed from the liquid nitrogen freezer and immediately immersed in a 37 °C water bath with gentle agitation for 3–5 min until thawed. After centrifugation at 1000 rpm for 5 min, the supernatant was replaced with 1 mL preheated MSC basal medium at 37 °C. The cell suspension was transferred to a 6 cm culture dish (123-17, ThermoFisher, Waltham, MA, USA) and mixed well with 2–3 mL basal medium. MSCs were cultured at 37 °C and passaged to multiple dishes when the confluency of cell reached 80–90%. When the confluency reached 80–90% again, the cells were passaged at a density of 20,000 cells/dish to several precoated confocal culture dishes (801002, NEST, China) (P2). Because stem cells cannot attach to a glass surface very well, the confocal culture dishes were pretreated with human plasma fibronectin solution (Fibronectin, FN, F0895-2MG, sigma, Saint Louis, MO, USA) before using. FN solution was added to cover the bottom of the confocal culture dish and incubated at 37 °C for 20 min. After incubation, FN solution was removed, and the dish was washed three times using PBS (10010023, ThermoFisher, Waltham, MA, USA) to remove residual FN solution. Subsequently, 1 mL basal medium was added into the confocal culture dish and multiple dishes of cells were placed in a 37 °C incubator for cultivation.

When the confluency of MSC reached 50%, osteogenic and adipogenic differentiation were induced separately in multiple dishes of MSCs. For the adipogenic group, the medium was changed to induction A medium for three days, followed by induction B medium for one day. This alternation pattern between A medium and B medium was repeated until Day 28. For the osteogenic group, the induction medium was replaced every three days until Day 28.

The cells were fixed after FLIM observation to preserve their original state. After the medium was removed from the culture dish, the dish was gently rinsed with 1 mL of PBS solution three times, 1 mL of 4% paraformaldehyde (I28800, ThermoFisher, Waltham, MA, USA) solution at room temperature was added for 20 min, the paraformaldehyde solution was removed, and the dish was rinsed three times with PBS.

### 2.3. FLIM Imaging and Processing

FLIM data of MSCs were acquired using a microscope equipped with a 60× oil immersion objective (N.A. = 1.40). The autofluorescence of NAD(P)H was excited using a femtosecond pulse laser (InSight X3 Dual, Spectra Physics, Milpitas, CA, USA) at a wavelength of 740 nm with a power of approximately 5 mW. The emission filter was set to 417–477 nm to ensure that the autofluorescence signal mainly originated from NAD(P)H. FLIM data were collected on Day 1, Day 7, Day 14, Day 21, and Day 28 after induction of MSCs.

FLIM data was fitted by a double-exponential function using LAS X Single Molecule Detection software (Leica, Weztlar, Germany). The average fluorescence lifetime was calculated as $\tau_{m}=a_{1}\tau_{1}+a_{2}\tau_{2}$, where τ_1_ and τ_2_ are the fluorescence lifetimes of free and protein-bound NAD(P)H, respectively, and a_1_ and a_2_ are the percentages of free and protein-bound NAD(P)H, respectively. Then the data were processed in batch using ImageJ software to obtain the FLIM images of a_2_ in the cells.

### 2.4. SRS Imaging

MSCs were fixed on Day 1, Day 7, Day 14, Day 21, and Day 28 after adipogenic induction for SRS imaging. The SRS imaging setup was reported previously [36]. The Stokes beam wavelength was 1045 nm with the power of 50 mW, and the pump beam wavelength was 804 nm with the power of 60 mW. A 60× oil immersion objective (N.A. = 1.40) and a bandpass filter (788.5–997.5 nm) were used for imaging. All samples were imaged in the lipid channel of 2850 cm^−1^.

### 2.5. ALP Staining of Osteogenic Differentiation and Imaging Methods

After FLIM observation of osteogenic induction cells at different time points, the cells were fixed and stained for quantitative analysis of osteogenic differentiation. The staining method involved removing the PBS in the culture dishes, adding 1 mL ALP staining solution (CTCC-JD002, Puhe Biology, Wuxi, China), incubating at room temperature for 90 min, removing the staining solution and rinsing three times with PBS. Bright-field imaging was performed using a microscope equipped with a 20× air objective (N.A. = 0.4) and a color CMOS camera (MV-CH089-10GC, Hikvision, Shanghai, China) with an exposure time of 20 ms.

### 2.6. Image Segmentation

The statistical results of NAD(P)H and lipid levels in cells showed the changes in cells from undifferentiated to differentiated (Appendix A). To study the cells at the single-cell level, image segmentation is necessary to obtain single-cell images. Due to the characteristics of MSCs, including a thin thickness after adhering to the substrate, high cell density, and unclear boundaries, common single-cell image segmentation techniques such as U-Net, SegNet, Mask-RCNN, and watershed segmentation cannot achieve satisfactory results. Therefore, a sliding window segmentation method was used to obtain single-cell images. The sliding window segmentation method was based on the work of previous research [37,38,39,40], and some modifications have been made to adapt to the segmentation of MSCs. The sliding window segmentation program was written in Spyder software in the Python 3.9 environment. A window size matrix was obtained by evaluating approximately 50 cell sizes under each condition for subsequent segmentation. The original image was cropped according to the elements in the window size matrix with step size of 10% window size [38]. After cropping, a closing operation was performed on all images to reduce small noise and holes in the images [41]. Following the closing operation, contour detection was applied to the images, and the detected contours were sorted according to their areas [42]. If cells were present in the cropped image, the largest contour corresponded to the cell and the second largest corresponded to the cell nucleus. After performing the convex hull of the second largest contour, the circularity and centroid were obtained, and when the circularity met threshold, the selected contour region could be determined as the cell nucleus [38]. Then, by comparing the centroid of the second largest contour with the center of the image, the presence of a single cell could be determined when the centroid was close to the image center [39,40]. By performing the above operations on all cropped images, single-cell images suitable for subsequent feature extraction and clustering analysis could be obtained. Cells with high density and unclear boundaries can be segmented by cropping.

### 2.7. Kmeans++ Algorithm

Due to the significant heterogeneity of MSCs during differentiation, the metabolic state and cell content distribution in each cell under the same conditions are not identical, making it impossible to assign a clear label to each cell. Therefore, an unsupervised machine learning method was used for cluster analysis. The K-means clustering algorithm, as an unsupervised machine learning algorithm, is widely used in various unsupervised clustering applications due to its simplicity, good clustering results, and strong interpretability. However, the choice of initial centroids in the K-means clustering process greatly affects the final clustering result. If the initial centroids are selected randomly, it may result in slow convergence and local optima [43]. Therefore, in this work, the optimized K-means++ algorithm was used for clustering calculations [43]. The K-means++ algorithm no longer adopts the approach of randomly selecting initial centroids. Instead, a data point k1 is randomly selected from the samples as the first centroid, and the distance between each remaining sample point and the initial centroid is calculated. The probability of selecting a point closer to the initial centroid as the second centroid is smaller. After selecting the second sample point as the second centroid, the distance between each remaining sample point and the second centroid is calculated again, and the third centroid is selected. This process is repeated until k centroids are selected, and the optimal clustering result is obtained after traversing all sample points. This approach effectively avoids the problem of local optima caused by initial centroids. In order to avoid the input image size being treated as an additional feature that affects the clustering result, 150–200 single-cell images at each time point during the induction of differentiation were used as the input, and the single-cell range was obtained by performing connected domain processing. Then, the features of the single-cell range were extracted, and the K-means++ algorithm was used to cluster the extracted features.

The feature extraction and clustering processes were implemented using Spyder (Python 3.9) software. The feature extraction and K-means++ clustering process is shown in Figure 1. The segmented images were subjected to adaptive thresholding, dilating, mean filtering, and finding the largest connected component to obtain the cell mask, which was then applied to the original image to obtain the single-cell image for feature extraction. Adaptive thresholding involves drawing a histogram of the image’s grayscale distribution, reading the peak value, and setting the pixel value greater than the peak value to 1 and less than the peak value to 0. To avoid losing too many pixels within the cell during binarization, the resulting binary image was dilated using a kernel size of 5. After dilation, a mean filter with a kernel size of 3 was applied to the image to remove background noise while smoothing the cell contour. Then, the largest connected component function was used to obtain a cell region mask, which was applied to the original single-cell image to obtain the final single-cell image for feature extraction after contour drawing.

Morphological features such as area, perimeter, and circularity were obtained using Python’s built-in functions from FLIM or SRS images. NAD(P)H lifetime features, including the peak value and full width at half maximum of the NAD(P)H a_2_ distribution histogram, were obtained by plotting the distribution histogram and reading the values from FLIM data. In the lipid content analysis, the average grayscale value was obtained by dividing the total grayscale value of the cellular region by the total number of pixels counted in SRS images.

The extracted features were normalized before determining the number of clusters. The number of clusters is determined by the elbow rule. More details on the principle and a more in-depth flowchart for determining the true number of clusters are provided in Appendix A and Appendix A. After determining the number of clusters, the feature data are finally input into the K-means++ algorithm for unsupervised clustering.

K-means++ clustering of FLIM data during adipogenic and osteogenic differentiation used area of cells, peak value, and full width at half maximum of NAD(P)H a_2_ distribution histogram as input features. Considering that NADH is a more effective feature for reflecting metabolic changes than cell morphology, morphological features such as perimeter and circularity of cells were added for clustering, but no significant improvement was observed in the results, and it increased the running time of program. K-means++ clustering of SRS data during adipogenic differentiation process used area, perimeter, circularity, and average grayscale value of cells as input features.

### 2.8. Statistical Analysis

A Z-test statistical analysis method suitable for large samples was selected because of the large sample size, and all statistical analysis was performed using Excel 2019.

## 3. Results

### 3.1. Identification of Adipogenic Differentiation Status

#### 3.1.1. FLIM Images and K-Means++ Clustering

Typical FLIM a_2_ images (the percentage of the protein-bound NAD(P)H) of MSCs in adipogenic differentiation from Day 1 to Day 28 are shown in Figure 2A, wherein red represents a_2_ below 30%, and green represents a_2_ above 30%. Approximately 150–200 cells were involved in the averaging for each condition. The statistical averaged peaks of the a_2_ distribution curve from Day 1 to Day 28 are shown in Figure 2B, where approximately 150–200 cells were involved in the averaging for each condition. The averaged peaks were 19% ± 4%, 20% ± 3%, 20% ± 5%, 18% ± 4%, and 19% ± 5%. It can be seen that there are no significant differences among them.

Considering that using a single feature for cluster analysis leads to issues such as model instability and sensitivity to outliers [44], multidimensional features were used in K-means++ clustering. The optimal number of clusters was three when inputting cell area, peak of a_2_ values, and half width at half maximum of the a_2_ curves from the FLIM data as features, as shown in Appendix A.

The more refined clustering results are shown in Figure 3, where (A)–(E) represent the clustering results on Day 1, Day 7, Day 14, Day 21, and Day 28, respectively. The black asterisks indicate the cluster centers, which are located at (0.63, 0.40, 0.64), (0.92, 0.32, 0.36), and (0.38, 0.31, 0.34) for the three clusters. The red, blue, and green colors represent the data points assigned to Cluster 1, Cluster 2, and Cluster 3, respectively. The clustering results for each time point of adipogenic induction are shown in Figure 2C. On Day 1 of adipogenic induction, 66% ± 3% of MSCs were distributed in Cluster 1, 9% ± 10% in Cluster 2, and 23% ± 11% in Cluster 3. On Day 7, 12% ± 7% of MSCs were distributed in Cluster 1, 54% ± 19% in Cluster 2, and 35% ± 25% in Cluster 3. On Day 14, 5% ± 5% of MSCs were distributed in Cluster 1, 60% ± 12% in Cluster 2, and 35% ± 17% in Cluster 3. On Day 21, 6% ± 3% of MSCs were distributed in Cluster 1, 50% ± 5% in Cluster 2, and 44% ± 7% in Cluster 3. On Day 28, 1% ± 3% of MSCs were distributed in Cluster 1, 32% ± 28% in Cluster 2, and 71% ± 22% in Cluster 3. It can be inferred that Cluster 1 represents undifferentiated cells, Cluster 2 represents cells in the process of differentiation, and Cluster 3 represents differentiated cells.

#### 3.1.2. SRS Images of Lipids and K-Means++ Clustering

Typical SRS images of lipids in MSCs in adipogenic differentiation from Day 1 to Day 28 are shown in Figure 4A. The number of lipid droplets increased gradually from Day 1 to Day 28 when MSCs were induced into adipogenic linages. Average grey values of lipid signals from Day 1 to Day 28, shown in Figure 4B, were 22,968 ± 4362, 25,978 ± 4711, 34,171 ± 7120, 33,767 ± 6512, and 34,219 ± 4408, respectively. The intensity of lipids increased from Day 1 to Day 14 and remained basically unchanged after Day 14.

Similar to the K-means++ clustering of FLIM images, multidimensional features were used as the input for K-means++ clustering of SRS images. The feature inputs included cell area, perimeter, circularity, and mean gray value in the SRS data. The optimal number of clusters was three, as shown in Appendix A. Due to the high dimensionality of the data, it was not possible to create an intuitive scatter plot, so only the proportions of cells in each cluster at each time point were calculated. Figure 4C shows the clustering results and data distribution of cells from Day 1 to Day 28 under adipogenic induction conditions. With four features as the input data, on Day 1 of adipogenic induction, 83% ± 5% of MSCs were distributed in Cluster 1, 1% ± 1% in Cluster 2, and 17% ± 4% in Cluster 3. On Day 7, 63% ± 8% of MSCs were distributed in Cluster 1, 2% ± 2% in Cluster 2, and 35% ± 8% in Cluster 3. On Day 14, 20% ± 9% of MSCs were distributed in Cluster 1, 46% ± 13% in Cluster 2, and 34% ± 14% in Cluster 3. On Day 21, 13% ± 6% of MSCs were distributed in Cluster 1, 37% ± 6% in Cluster 2, and 50% ± 1% in Cluster 3. On Day 28, 4% ± 2% of MSCs were distributed in Cluster 1, 41% ± 6% in Cluster 2, and 55% ± 4% in Cluster 3. It can be inferred that Cluster 1 represents undifferentiated cells, Cluster 2 represents differentiating cells, and Cluster 3 represents differentiated cells.

The distribution patterns in Figure 2C and Figure 4C are generally consistent, confirming that the combination of the two label-free microscopy techniques can help obtain more comprehensive information on stem cell differentiation. The main difference lies in the classification of differentiating cells on Day 7. This may be due to the fact that FLIM and SRS, respectively, observe differences in metabolism and lipid contents. From the perspective of cell differentiation, metabolic changes occur earlier than the accumulation of lipid contents. Therefore, more cells were determined to be undergoing differentiation using FLIM on Day 7. These cells had begun differentiation but had not accumulated enough lipid contents, so they were evaluated as undifferentiated using SRS on Day 7. This also reflects the sensitivity and observation features of the two techniques.

### 3.2. Identification of Osteogenic Differentiation Status

Since the unsupervised K-means++ clustering method using multiple features as the input was found reliable for determining the differentiation status of individual cells during adipogenic induction, it was tested for its ability to discriminate the differentiation status of individual cells during osteogenic induction. During the process of osteogenic differentiation of MSCs, hydroxyapatite (HA) gradually accumulates in the extracellular matrix. HA has a strong SRS signal, but this signal cannot accurately characterize the differentiation state of individual cells. Therefore, only FLIM data analysis was conducted, and the results were validated by ALP staining.

#### 3.2.1. FLIM Images and K-Means++ Clustering

Typical FLIM a_2_ images of MSCs induced to undergo osteogenic differentiation from Day 1 to Day 28 are shown in Figure 5A. The statistical averaged peaks of the a_2_ distribution curve from Day 1 to Day 28 are shown in Figure 5B. Approximately 150–200 cells were involved in the averaging for each condition. The average peaks were 22% ± 2%, 25% ± 3%, 25% ± 3%, 27% ± 4%, and 26% ± 2%, respectively. There were no significant differences among them except the data of Day 28. For cluster analysis of FLIM images of MSCs induced to undergo osteogenic differentiation, the multidimensional features were the same as those of MSCs induced to undergo adipogenic differentiation, and the optimal number of clusters was three, as shown in Appendix A.

The detailed clustering results are shown in Figure 6, where (A)–(E) represent the clustering results of Day 1 to Day 28. The black asterisks represent the cluster centers with coordinates of (0.20, 0.60, 0.51), (0.18, 0.71, 0.69), and (0.32, 0.72, 0.52). The red dots represent the data points assigned to Cluster 1, the blue dots represent those assigned to Cluster 2, and the green dots represent those assigned to Cluster 3. Figure 5C shows the K-means++ clustering results with three features of induced osteogenic differentiation used as the input from Day 1 to Day 28. On Day 1, 87% ± 7% of MSCs were distributed in Cluster 1, 2% ± 0.1% in Cluster 2, and 11% ± 7% in Cluster 3. On Day 7, 23% ± 10% of MSCs were distributed in Cluster 1, 69% ± 12% in Cluster 2, and 8% ± 4% in Cluster 3. On Day 14, 17% ± 6% of MSCs were distributed in Cluster 1, 60% ± 6% in Cluster 2, and 24% ± 5% in Cluster 3. On Day 21, 25% ± 8% of MSCs were distributed in Cluster 1, 47% ± 15% in Cluster 2, and 31% ± 13% in Cluster 3. On Day 28, 18% ± 11% of MSCs were distributed in Cluster 1, 10% ± 2% in Cluster 2, and 72% ± 13% in Cluster 3. Based on the above results, it can be inferred that Cluster 1 represents undifferentiated cells, Cluster 2 represents cells in the process of differentiation, and Cluster 3 represents fully differentiated cells.

#### 3.2.2. ALP Staining of Osteogenic Differentiation

To verify whether the K-means++ clustering results were consistent with the actual differentiation statuses, the differentiation statuses of cells were evaluated using chemical staining. ALP is a highly active enzyme during osteogenic differentiation and is commonly used to characterize osteogenic differentiation. Therefore, ALP staining solution was used to stain induced cells from Day 1 to Day 28. The staining results and the statistics of differentiated cells are shown in Figure 7, with approximately 300 cells counted at each time point.

As shown in Figure 7A, the percentage of blue-stained cells indicating ALP activity in the induced cells increased gradually with the induction time, and the intensity of the staining also increased. Since it was difficult to quantitatively analyze the intensity of blue staining in each cell, the cells were only divided into two categories: stained and unstained. The percentage of stained cells was then calculated and plotted in Figure 7B for each time point, based on the staining results of approximately 300 cells. The percentage of stained cells was only 9% ± 0.4% on the first day, but it gradually increased to 78% ± 3%, 86% ± 1%, 88% ± 2%, and 89% ± 1% on days 7, 14, 21, and 28, respectively. Since ALP is highly expressed during osteogenic differentiation, the stained cells should include both differentiating and differentiated cells, which correspond to the cells in Clusters 2 and 3 in Figure 6. The results of summing the cells of Clusters 2 and 3 in Figure 2 are shown in Figure 7C, which are consistent with the results in Figure 7B. This confirms the reliability of the conclusions in Figure 6C and Figure 7.

## 4. Discussion

The induction media for osteogenic and adipogenic differentiation of MSCs have been widely utilized in the process of MSCs differentiation [16,45,46]. ALP is a biomarker for bone formation and was utilized by Guo et al. to stain MSCs during a 28-day osteogenic differentiation process, demonstrating the differentiation of MSCs into osteoblasts [16]. This observation is consistent with the findings presented in Figure 6 and Figure 7. Other studies have also employed induction media to induce osteogenic and adipogenic differentiation of MSCs for a duration of 21 days, confirming the successful differentiation into osteoblasts and adipocytes through staining with alizarin red and oil red O [45,46]. ALP or alizarin red staining can demonstrate that MSCs can differentiate into osteoblasts following 21 days of induction [16,46]. Therefore, it is possible to discriminate differentiated cells at an earlier stage using FLIM. SRS imaging, as a highly sensitive detection technique for lipid signals, has been extensively utilized in the detection of adipocytes [47,48,49] and can be an alternative to oil red O staining. Therefore, utilizing ALP staining for the detection of osteoblasts and SRS for the detection of adipocytes can show reliable changes in the differentiation levels of MSCs.

During the differentiation process of MSCs, complex metabolic changes and cell content changes occur within individual cells. During the induction of adipogenic differentiation, the metabolic changes are reflected in the shift from glycolysis to oxidative phosphorylation, while the cell content changes are reflected in the gradual accumulation of lipid droplets within the cells. During the induction of osteogenic differentiation, the energy metabolic changes are similar to adipogenic cells, with a shift from glycolysis to oxidative phosphorylation [29]. However, the cell content changes are reflected in the high expression of alkaline phosphatase within the cells. The metabolic and cell content changes can be depicted using FLIM and SRS, respectively.

The changes in intracellular FLIM a_2_ data are closely related to the metabolic state of the cell [50]. Therefore, the metabolic states of cells undergoing adipogenic and osteogenic differentiation were characterized by collecting data using FLIM imaging. The metabolic changes of differentiated MSCs are intuitively shown in Figure 2A and Figure 5A by the transition of a_2_ in MSCs from red to green. Since metabolic changes during cell differentiation are highly complex and heterogeneous, with vastly different metabolic states among different cells under the same conditions, using statistical averaging methods to determine differentiation status is not suitable for characterizing the cellular state and is inadequate for analyzing large amounts of highly heterogeneous cell data; for example, Figure 2B and Figure 5B show that the results of a_2_ had no significant differences. Manual analysis is time-consuming and makes it difficult to ensure accuracy, whereas using an unsupervised machine learning method to perform cell classification analysis enables analysis of individual cells and ensures robust data. In this article, the K-means++ algorithm, which is simple to use and produces good clustering results, was employed to perform cluster analysis on the FLIM data obtained from approximately 1000 cells. From the results in Appendix A, it can be seen that the best cluster number for all processes of clustering is three, which is consistent with the states of cell differentiation, corresponding to the undifferentiated (Cluster 1), differentiating (Cluster 2), and differentiated (Cluster 3) states [51,52].

Figure 2C shows the statistical distribution of cell states during the induction of adipogenic differentiation from Day 1 to Day 28. The more detailed clustering results are illustrated in Figure 3. The percentage of undifferentiated cells gradually decreased, while the percentages of differentiating and differentiated cells increased during the process of adipogenic differentiation from Day 1 to Day 28, consistent with the findings of Mehta et al. [53]. The distribution of cells in different states during adipogenic differentiation was similar to that of osteogenesis. In the early stage of induction, the metabolic states of different cells varied greatly, resulting in a relatively scattered distribution of data within each cluster. However, in the late stage of induction, most cells had either completed or almost completed differentiation, and their metabolic states tended to be consistent, leading to a highly concentrated distribution of data within each cluster. SRS imaging, a technique that is highly sensitive to lipid signals, can accurately detect the gradually increasing lipid signals within cells during the induction of differentiation. Figure 4A shows that lipid droplets in cells accumulated gradually from Day 1 to Day 28. Although the mean grey value of cells increased from Day 7 to Day 28, it could not reveal the heterogeneity of lipid content on the single-cell level. Thus, the statistical distribution of clustering results from SRS data was presented in Figure 4C, showing that the percentage of cells in Cluster 1 decreased gradually, whereas the percentages of cells in Clusters 2 and 3 increased gradually. The clustering results of SRS data further validated the clustering results obtained from FLIM data and demonstrated that the unsupervised K-means++ clustering method, which combines FLIM and SRS data, can be used to classify undifferentiated, differentiating, and differentiated cells at the single-cell level without labeling, with high efficiency and accuracy, and without the problems caused by ignoring cell heterogeneity that may occur in simple statistical averaging methods.

The clustering results based on three features of the osteogenic differentiation process are shown in Figure 5C and Figure 6. The proportions of clustered cells in each cluster from Day 1 to Day 28 corresponded to the process of cell differentiation, with a gradual decrease in undifferentiated cells and an increase in differentiated cells. In addition, the data distribution was more dispersed in the early stage of differentiation and more concentrated in the late stage, which was due to the differences in metabolic states among different cells at the early stage of induction, while most cells were close to completion or had completed differentiation at the late stage, resulting in a more consistent metabolic state. These findings were consistent with the findings of a previous report [22]. The ALP staining schematic and statistical results from Day 1 to Day 28 of induced differentiation shown in Figure 7 further confirmed that the proportion of differentiated cells increased with the induction time. Since ALP was highly expressed in stem cells after the start of differentiation [54], and the degree of staining in individual cells could not be quantitatively analyzed, the staining results were divided into unstained (undifferentiated) and stained (differentiating or differentiated). The sum of the proportions of differentiating and differentiated cells in the clustering results shown in Figure 7B was also close to the sum of the proportions of stained cells in Figure 7C, and the large error bars in the clustering results were possibly due to the heterogeneity of the cells, and the possibility that the differentiation states of cells in different regions could vary. In addition, MSCs are heterogeneous, which exhibit spontaneous differentiation towards osteogenic and adipogenic lineages during proliferation under basic culture conditions, as indicated by increased levels of alkaline phosphatase and lipid expression, respectively [55,56]. Due to the approximately 6-day period required for recovery, expansion, and seeding of cells in confocal culture dishes for subsequent imaging before induction of differentiation, a small proportion of cells may have undergone spontaneous differentiation, resulting in the presence of differentiated cells on Day 1 of induction (Cluster 3). In future studies, efforts should be made to minimize the duration of time before induction in order to reduce the impact of spontaneous differentiation.

These results indicated that different MSCs induced under different conditions have little heterogeneity in their differentiation status and exhibit similar proportions of each cluster at different time points. By combining machine learning analysis methods, high-throughput, label-free classification of osteogenic MSCs at the single-cell level was achieved.

## 5. Conclusions

The heterogeneity of differentiation states in MSCs during adipogenic and osteogenic differentiation was investigated using an analysis approach combining FLIM or SRS with the unsupervised K-means++ clustering algorithm. The classification of undifferentiated, differentiating, and differentiated MSCs into adipogenic and osteogenic lineages at the single-cell level was achieved noninvasively and efficiently without labeling. The proposed analysis approach provides a convenient tool for metabolic studies of cell differentiation processes and enables the recycling and reuse of differentiated cells. Furthermore, this approach offers a rapid and nondestructive detection method for the application of stem cells in tissue engineering and clinical therapy, which may promote further development of stem cell applications in these fields.

## Figures and Tables

**Figure 1 cells-12-01524-f001:**
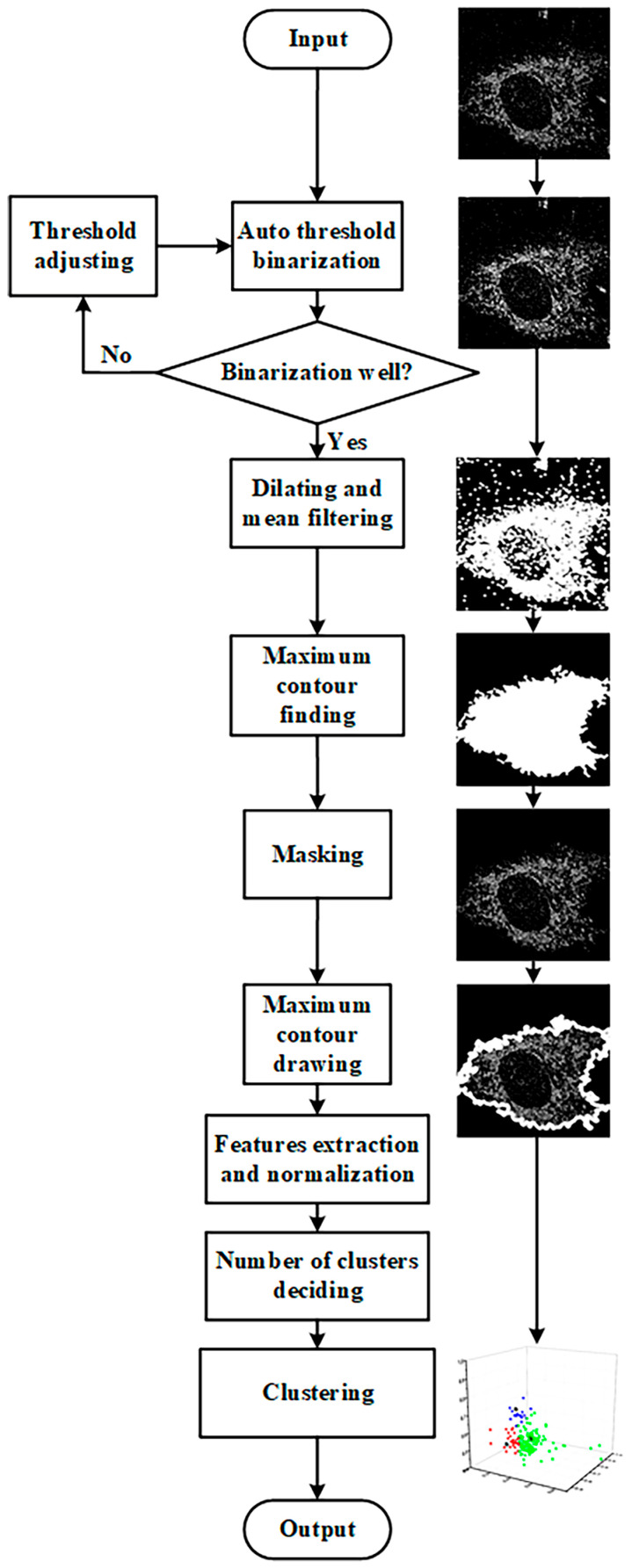
Feature extraction and K-means++ clustering process.

**Figure 2 cells-12-01524-f002:**
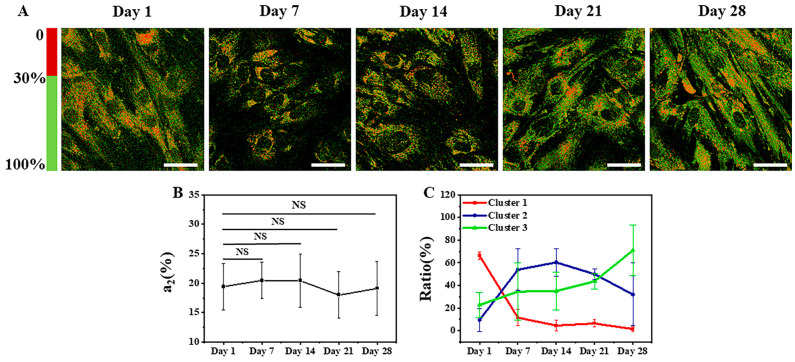
Typical FLIM images and K-means++ clustering results with three features input of induced adipogenic differentiation from Day 1 to Day 28. (**A**) FLIM a_2_ images (percentage of protein-bound NAD(P)H). Scale bar: 50 μm. (**B**) The averaged peaks of the a_2_ distribution curve at different times. NS: no significant difference. (**C**) The ratios of the cells in the three clusters.

**Figure 3 cells-12-01524-f003:**
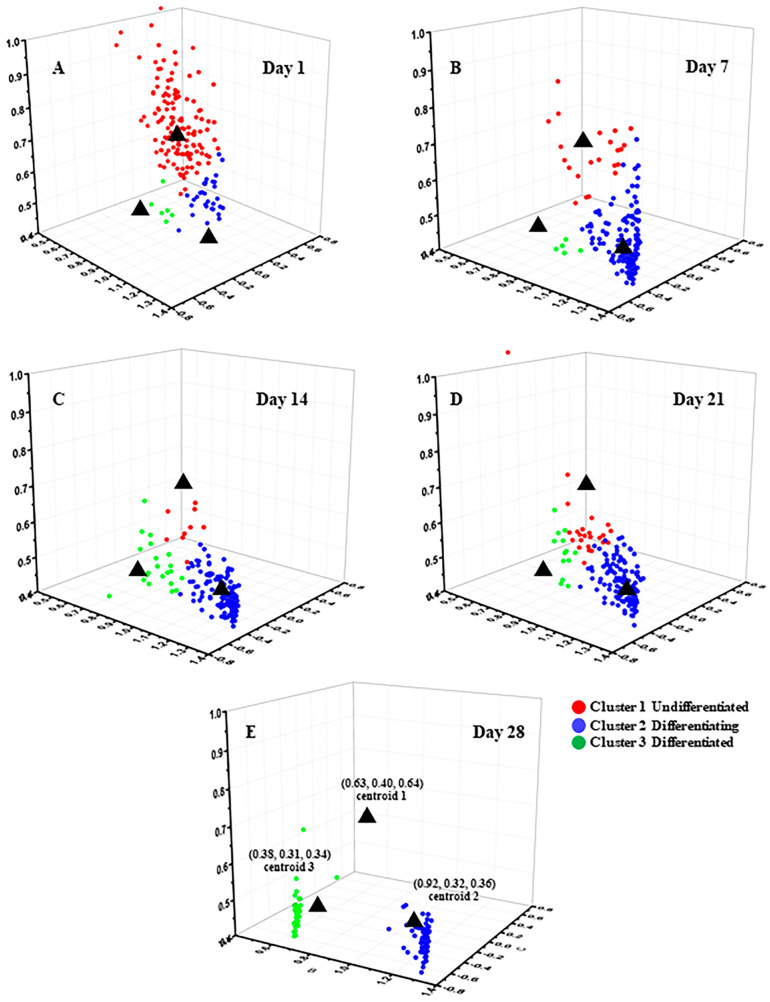
K-means++ clustering distribution with three features input of induced adipogenic differentiation from Day 1 to Day 28. (**A**) Day 1. (**B**) Day 7. (**C**) Day 14. (**D**) Day 21. (**E**) Day 28.

**Figure 4 cells-12-01524-f004:**
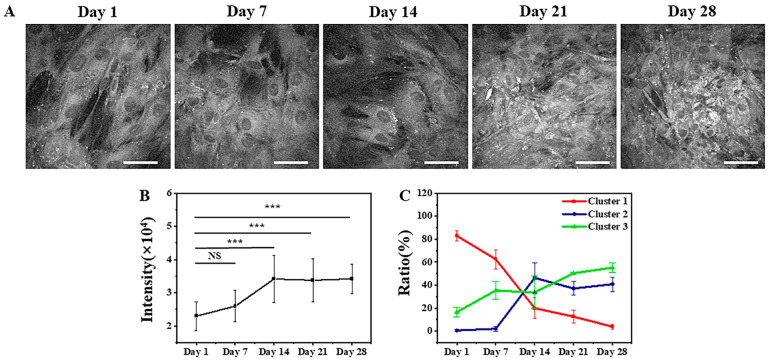
Typical SRS images of lipids in MSCs and K-means++ clustering results with four features input of induced adipogenic differentiation from Day 1 to Day 28. (**A**) SRS images. Scale bar: 50 μm. (**B**) The averaged grey values of lipid signals at different times. NS: no significant difference, ***: *p* < 0.001. (**C**) The ratios of the cells in the three clusters.

**Figure 5 cells-12-01524-f005:**
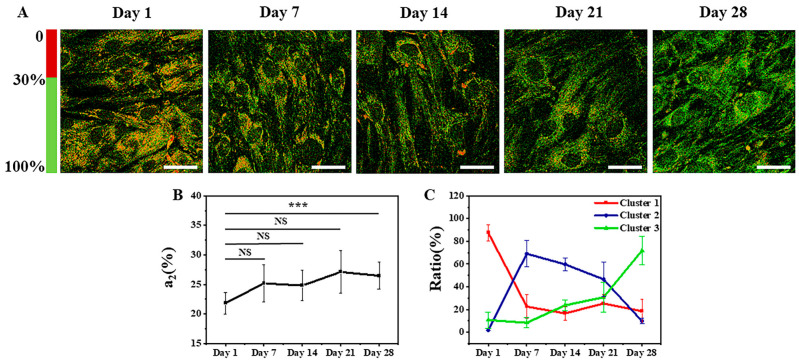
Typical FLIM images and K-means++ clustering results with three features input of induced osteogenic differentiation from Day 1 to Day 28. (**A**) FLIM a_2_ images (percentage of protein-bound NAD(P)H). Scale bar: 50 μm. (**B**) The averaged peaks of the a_2_ distribution curve at different times. NS: no significant difference. NS: no significant difference, ***: *p* < 0.001. (**C**) The ratio of the cells in the three clusters.

**Figure 6 cells-12-01524-f006:**
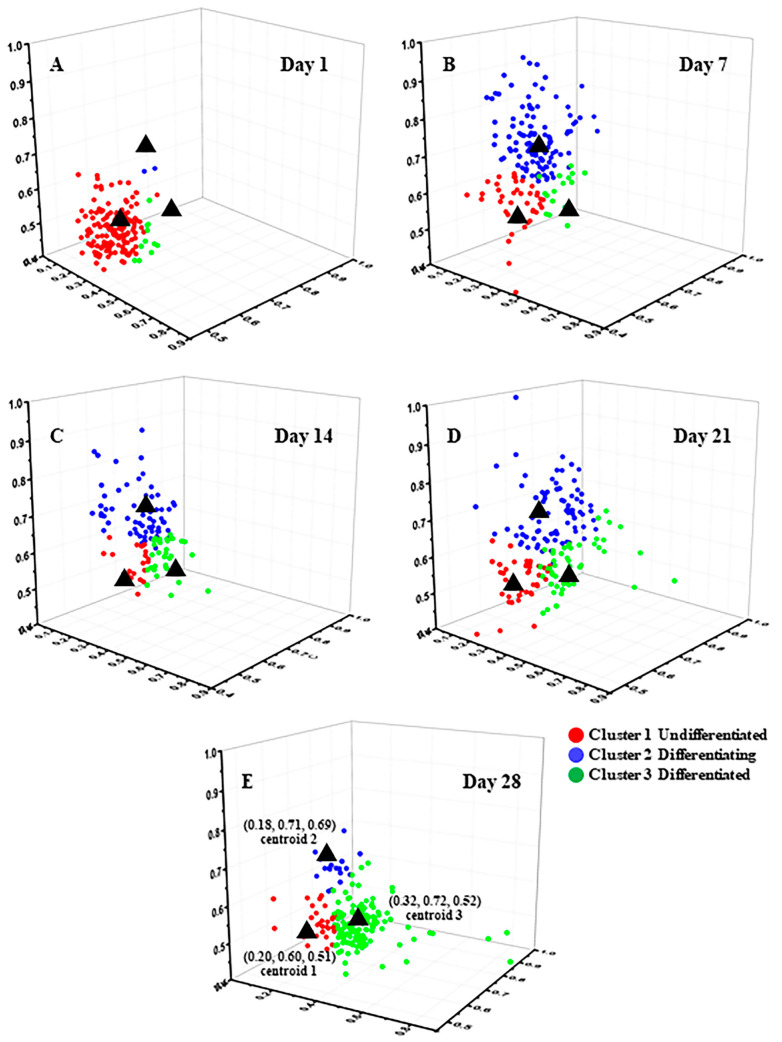
K-means++ clustering distribution with three features input of induced osteogenic differentiation from Day 1 to Day 28. (**A**) Day 1. (**B**) Day 7. (**C**) Day 14. (**D**) Day 21. (**E**) Day 28.

**Figure 7 cells-12-01524-f007:**
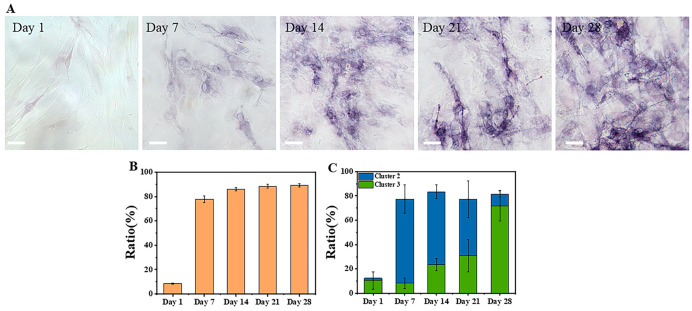
Results of ALP staining and differentiated ratios of MSCs induced to undergo osteogenic differentiation from Day 1 to Day 28. (**A**) ALP staining images of MSCs. Scale bar: 50 μm. (**B**) Ratios of stained cells. (**C**) Sum of Cluster 2 and Cluster 3 in K-means++ clustering result.

## Data Availability

All data that support the findings of this study are available from the corresponding author upon reasonable request.

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
