# Peer review of "Evaluating Differentiation Status of Mesenchymal Stem Cells by Label-Free Microscopy System and Machine Learning"

_cells, 2023, doi:10.3390/cells12111524_

Round 1

Reviewer 1 Report

The paper titled: “Evaluating Differentiation Status of Mesenchymal Stem Cells by Label-Free Microscopy System and Machine Learning “ Yawei Kong at al. raises a very interesting, important  and current topic .

Despite the fact that the subject is very interesting, the results are unfortunately poorly documented.

If the authors want to prove that their algorithm is correct, the microscopic method should be accompanied by a classic assessment of the NADH level after cell lysis to assess whether both methods actually show reliable changes in the metabolism.

Until such validation is carried out (change of NADP at different levels of cell differentiation), further algorithms have no confirmation.

The language needs extensive correction. "MSCs  were resuscitated from liquid nitrogen" is one of the examples.

However, the manuscript was preapared by people for whom English is not a native language, so this fact does not affect the value of the work.

Reviewer 2 Report

The authors conducted a study to evaluate the differentiation of hMSCs induced in a specific cell culture medium using label-free microscopic images and machine learning. They found that the model can sensitively analyze individual cell differentiation status. However, several major points should be considered, including:

1. The "Materials and Methods" section is unclear and could be improved to better explain the features and how they were extracted. The authors mention "multidimensional features" and show three or four features in some figures, but it is unclear which features they are referring to (e.g. shape, etc..).

2. The feature extraction and K-means++ clustering process shown in Figure 1 section are not clear enough, and the authors could improve the scheme's explanation. Also, “Kmeans++ algorithm” section finish with the sentence “More details on the principle and a more in-depth flowchart are provided in Supplementary Materials and Figure S1”. None supplementary Materials is provided. Some questions are: the data extracted (features) were standardized/scaled? Which method is used in selecting number of clusters? Etc….

3. Regarding the sliding window segmentation method used to obtain single-cell images, there is not enough reference or detail on the process or software used. An example processed image could clarify the segmentation of high cell density with unclear boundaries, which is a crucial point.

4. The article includes statistical analysis; however, the authors do not describe the analysis in the methods section or how it was performed.

5. The figures are not clear enough to read, and for Cluster 3 in the figures, it is unclear how differentiated cells can be observed after only one day. The authors need to clarify these sentences appropriately.
